# Survey on Limnic Gastropods: Relationships between Human Health and Conservation

**DOI:** 10.3390/pathogens11121533

**Published:** 2022-12-13

**Authors:** Paulo R. S. Coelho, Fabricio T. O. Ker, Amanda D. Araujo, Hudson A. Pinto, Deborah A. Negrão-Corrêa, Roberta L. Caldeira, Stefan M. Geiger

**Affiliations:** 1Laboratory of Intestinal Helminthiasis, Department of Parasitology, Institute of Biological Sciences, Federal University of Minas Gerais, Belo Horizonte, Brazil; 2Laboratory of Epidemiology of Infectious and Parasitic Diseases, Department of Parasitology, Institute of Biological Sciences, Federal University of Minas Gerais, Belo Horizonte, Brazil; 3Research Group on Helminthology and Medical Malacology, René Rachou Institute, Oswaldo Cruz Foundation, Belo Horizonte, Brazil; 4Laboratory Trematoda Biology, Department of Parasitology, Institute of Biological Sciences, Federal University of Minas Gerais, Belo Horizonte, Brazil; 5Laboratory of Immunohelminthology and Schistosomiasis, Department of Parasitology, Institute of Biological Sciences, Federal University of Minas Gerais, Belo Horizonte, Brazil

**Keywords:** malacofauna, freshwater snails, diversity, environmental characterization, trematodes, cercariae

## Abstract

The present work aimed to study ecological aspects related to the distribution pattern of medically important and native freshwater mollusks, found in a rural municipality in the state of Minas Gerais, Brazil. Malacological captures were carried out in aquatic environments (lentic and lotic) from 46 locations between October 2018 and September 2019. The collected specimens were subjected to taxonomic identification and evaluation for infection with trematode larvae. Qualitative data were used to analyze the similarity and the odds ratios between the environmental variables. In total, 1125 specimens were sampled, belonging to the following species: *Biomphalaria glabrata, B. tenagophila, B. straminea, B. kuhniana, B. cousini, Biomphalaria* sp., and *Drepanotrema cimex* (Planorbidae), *Stenophysa marmorata* (Physidae), *Omalonyx* sp. (Succineidae), *Pseudosuccinea columella* (Lymnaeidae), and *Pomacea* sp. (Ampullaridae). Echinostome, strigeocercaria, and xiphidiocercaria types of larval trematodes were detected in *S. marmorata* and *D. cimex*. Of note was the similarity in the distribution of *S. marmorata*, a supposedly endangered species, with that of the medically important *Biomphalaria* species, with the two sharing environments. This complex scenario led us to reflect on and discuss the need for the control of important intermediate hosts, as well as the conservation of endangered species. This relevant issue has not yet been discussed in detail, in Brazil or in other countries that recommend snail control.

## 1. Introduction

It is estimated that about 4000 species of freshwater gastropods exist in the world [1]. In many ecosystems, these mollusks are important food sources [2] and also contribute to nutrient recycling [3]. As such, freshwater snails are considered excellent bioindicators of environmental quality, present sufficient abundance, and have the ability to adapt to different environments [4]. More importantly, mollusks can act as intermediate hosts of parasitic helminths, including several species of medical or veterinary importance worldwide [5].

Despite the great diversity and importance of these organisms, their distribution is not completely known in most areas of the globe. Thus, the realization of inventories of limnic gastropods is of paramount importance to promote knowledge on the diversity, biology, distribution, and dispersion, and on ecological relationships between populations and communities [6,7,8,9,10]. In this context, surveys can also expand the understanding of medically important species and/or native species.

The related malacological records for diverse freshwater environments, in most cases, have been dedicated to species associated with trematodes of medical and veterinary importance [11,12,13,14], especially the studies on schistosomiasis and fasciolosis. On the other hand, most of the other trematode species reported have complex biological cycles and intermediate mollusk hosts that are still unknown or are described merely in part [11,12].

For these invertebrates, little emphasis has been placed on populations and communities in research designs, and there are only a few scientists who are specialists in this field. However, snails have already been targeted in intervention programs in the African continent, in Central and South America, as well as in East Asia, but there is still much to be discussed about these interventions, before a general recommendation for nationwide control programs can be launched [15,16,17]. An approach that has grown in recent years involves ecological analyses to understand the spatiality of snails and the characterization of their habitats, in order to control diseases caused by trematode parasites [18].

Thus, the present research aimed to study the composition and distribution of freshwater mollusks in the municipality of Alvorada de Minas, State of Minas Gerais, Brazil. Within this perspective, the identification of the collected specimens, the evaluation of the associated trematodes, and the similarity and relationship between shared environments by the different snail species were evaluated.

## 2. Materials and Methods

The municipality of Alvorada de Minas (18°43′7″ S 43°22′5″ W) is located in the mesoregion of the Metropolitan area of Belo Horizonte and microregion of Conceição do Mato Dentro, 210 km north of the capital of Minas Gerais; it is part of the “Estrada Real” tourist route. The main tributary is the Rio do Peixe river, which is a sub-basin of the Rio Doce River, one of the main hydrographic basins of Minas Gerais.

Freshwater mollusks were collected at 17 sampling sites from 46 investigated locations, representative of lentic and lotic ecosystems of the municipality. The malacological survey was carried out on five occasions: October (spring) in 2018; February (summer), March and May (autumn), and September (winter) in 2019. Snail captures were done by three trained collectors with a sampling effort of 20 min at each sampling point, according to the technique of Olivier and Schneiderman [19], adapted for qualitative analyses. Scoops for snail capture were used, as recommended by the Brazilian Schistosomiasis Control Program (Ministry of Health, 2007). Georeferencing of the collection sites was made with the aid of a portable GPS device (GARMIN GPSMAP 62S).

For the identification of mollusks, morphological data were used that followed the conchological criteria based on specific identification for each species using the references: *P. columella* (the intermediate hosts of *Fasciola hepatica*) [20,21]; *Pomacea* sp.; *Physa* sp.; *Drepanotrema* sp. and *Omalonyx* sp. [22,23]; *Stenophysa* sp. [24]. For the genus *Biomphalaria* (the intermediate hosts of *Schistosoma mansoni*), morphological analyses were performed [25,26] and complemented by molecular methods [27,28]. The mollusks obtained were photographed (LEICA DFC450), and voucher specimens were deposited in the Medical Malacology Collection (CMM) of the René Rachou Institute (IRR) of the Oswaldo Cruz Foundation (Fiocruz). The field procedures and collections were authorized by the Chico Mendes Institute for Biodiversity Conservation (SISBIO registration number nº 68627-1).

For the active search of shed cercariae, collected snails were individualized and incubated in small, nonsterile tissue culture plates (24-well, Costar) with filtered and dechlorinated water, and exposed to light of incandescent lamps (60 W) for 4 h [29], except for *Omalonyx* sp.. The visualization of the cercariae was done with a stereoscopic microscope. Obtained larvae were photographed (LEICA ICC50 HD), and the identification of cercariae was based on morphological criteria, according to [11,30]. Negative snails were crushed to confirm the absence of other larval stages.

For statistical analyses, the frequencies obtained from the categorical data of the environments (lotic and lentic) and the presence of snails were compared using the chi-square test.

Geocoded-based cartography, projection/datum with the SIRGAS 2000 UTM Zone 23S, was used for the spatial distribution of the mollusks. In the QGis, the databases with the municipal limits and census sectors from the Brazilian Institute of Geography and Statistics (IBGE) and hydrographic mesh from the Mineiro Institute of Water Management (IGAM) were used.

Due to the lack of water management in the municipality, the study area was strategically divided into hydrographic units (HU1 to HU12), considering the characteristics of relief, drainage, land use, the potential for mollusks to spread, and the distribution of sample points. Each hydrographic unit was composed of a main water body (brook) and its upstream tributaries (brook, ponds, lakes, or springs), regardless of whether it flows directly into the Peixe River or into another water body, e.g., HU1, HU2, and HU10. The Peixe River and its small tributaries were considered as a separate hydrographic unit (HU4). Moreover, two hydrographic units (HU1 and HU2) are affluents of the Peixe River, although they are in a neighboring municipality.

The qualitative similarity between the composition of the mollusk specimens and the relationship between the hydrographic units was observed through a simple correlation based on the Jaccard index, which considers the number of species common between two areas (a) and the number of exclusive species from each area (b, c) [31]:J = 100a/(a + b + c)

Based on this index, a bidirectional analysis was elaborated in the form of a dendrogram, where the grouping is made from the arithmetic mean of the elements and the values of the ordinates express the similarity relations between the objects indicated in the abscissa [32]. For this analysis the software Paleontological statistics (PAST, version 4.10) was used.

In order to elucidate whether there was a relationship between the presence of *B. glabrata* and the presence of other mollusks, a logistic regression method was used. Logistic regression fits into the general framework of two generalized linear models (GLM) and can be used to analyze the relationship between a binary response variable and one or more explanatory variables [33], from the “presence/absence response curve” of a species [34]. The variables were dichotomized according to the presence or absence of species at the sampling point. Differences with a *p*-value ≤ 0.05 were considered statistically significant. Subsequently, the models were validated through the percentage of correct classification, based on the ROC curve analysis. This statistical analysis was performed using the STATA version 15.1 program (STATA Corporation, College Station, TX, USA).

## 3. Results

The malacological investigation resulted in the registration of different families, frequently identified in water collections of southeastern Brazil [35,36], namely, Planorbidae, Physidae, Succineidae, Lymnaeidae, and Ampullaridae. A total of 1125 specimens of gastropods from 11 species and five families were identified in the water resources of the municipality of Alvorada de Minas. Among the collected snails, 224 were identified as *Stenophysa marmorata* (Guilding, 1828) (Physidae); 66 as *Drepanotrema cimex* (Moricand, 1839) (Planorbidae); 44 as *Omalonyx* d’Orbigny, 1837 (Succinidae); 16 as *Pseudosuccinea columella* (Say, 1817) (Lymnaeidae); 8 as *Pomacea* sp. (Ampullaridae); 767 as *Biomphalaria* (Planorbidae). In the latter case, more detailed data were previously published [37], with the following species identified: *Biomphalaria glabrata* (Say, 1818), *Biomphalaria tenagophila* (d’Orbigny, 1835), *Biomphalaria straminea* (Dunker, 1848), *Biomphalaria kuhniana* (Clessin, 1883), and *Biomphalaria cousini* (Paraense, 1966) (Figure 1 and Figure 2).

Regarding the intermediate hosts of *S. mansoni*, causative agent of human intestinal schistosomiasis, previous work identified the presence of *B. glabrata* and *B. straminea* in Alvorada de Minas between 2012 and 2014 [38]. In the more recent field work carried out by our group [37], the presence of three main intermediate hosts was confirmed, including the first record of *B. tenagophila* in the municipality, together with an identification of possible risk areas for schistosomiasis transmission.

In the present survey, the occurrence of *P. columella*, an intermediate host species of *F. hepatica*, was recorded for the first time within the borders of the municipality. Fasciolosis is a parasitosis with worldwide distribution in ruminants and it is considered an important veterinary disease, due to considerable economic losses in livestock production [39,40,41]. This malacological record is important since Alvorada de Minas is a region that depends on agriculture and cattle breeding as the main economic activities. Until recently, the occurrence of *P. columella* had been documented in 40 municipalities in Minas Gerais [35,36,37,38,39,40,41,42], but had not yet been recorded for Alvorada de Minas. Since fasciolosis is a zoonosis that can also affect humans and can cause serious clinical manifestations in chronically infected individuals [39], the present record might be also of interest in the view of a One Health perspective [35,43]. As such, cases of human fasciolosis have been reported in Brazil in studies from the states of Bahia, Minas Gerais, Mato Grosso do Sul, Paraná, Rio de Janeiro, Rio Grande do Sul, Santa Catarina, and from São Paulo [44]. In total, 62 human cases have been confirmed in the past [45].

Another interesting finding of the study was the presence of *S. marmorata*, corresponding to the species *Physa marmorata* “Name uncertain”, according to others [25,46,47]. Although the taxonomy presented by Paraense (1986) is outdated, the species *Ph. marmorata* was included in the 2018 and 2021 list of threatened species by the Ministry of the Environment (MMA) and the Chico Mendes Institute for Biodiversity Conservation (ICMBio), where it was classified as a “vulnerable” species. Despite being widely distributed throughout the tropical regions of Central America and South America [22], a reduction in the registration of populations was detected during the last decade due to environmental degradation and competition with exotic species [48,49,50,51]. In the present survey, this species was identified in 13 of the 17 locations where freshwater mollusks were found.

The investigation of trematode larvae associated with the collected mollusks revealed the presence of larval stages identified as echinostome and xiphidiocercaria types of cercariae, both shed by *S. marmorata* (infection rate: 0.89%). In addition, strigeiocercaria were found in two specimens of *D. cimex* from different localities (Figure 3). These parasites usually parasitize wild amphibians, reptiles, or birds [11]^,^ and further studies are required to advance the identification of these trematodes.

The aquatic collections were classified as lentic (stagnant or low water flow, especially lakes and ponds) and lotic (running water, streams, waterfalls, and rivers). Forty-five environments were evaluated, classified into 29 (64.4%) lentic environments and 16 (35.6%) lotic environments. Collection point 41 was not evaluated because it was an animal drinking trough, although it served as an excellent snail breeding site. The frequency in lentic environments was significantly higher (15/29; 51.75%) when compared to lotic environments (1/16; 6.2%) (*p* = 0.02338) (Table 1). At the collection sites, there was no record of exotic gastropods.

The sampled and described environments corroborated with data from the literature, where different authors reported that gastropods prefer lentic environments and have benthic habits [52,53], although it was also shown that they can colonize lotic environments [13]. Other characteristics such as the absence of riparian vegetation (only small grasses and aquatic vegetation), accentuated erosion, absent odor, visible changes of urban origin, and partially transparent waters were predominant at the sampling points (Figure 4).

Among the twelve hydrographic units of the municipality, there was no record of mollusks in four units (HU2, HU6, HU8, and HU10). Gastropods were recorded in eight hydrographic units (HU1, HU3, HU4, HU5, HU7, HU9, HU11, and HU12), with trematode larvae found in two of them (HU5 and HU9) (Figure 5). Based on Jaccard’s analysis Appendix A, bidirectional clustering was observed based on the similarity calculated for the distribution of gastropods. Hydrographic units HU3 and HU12 had a high similarity value, and HU7 and HU11 had low similarity. The low similarity of these environments is due to the occurrence of *Omalonyx* sp. and *B. cousin,* which were not recorded in other locations, and the absence of the species *B. glabrata* and/or *S. marmorata*. Analyses carried out from the gastropod perspective indicated that 10 of the 11 specimens showed similarity to each other with shared environments, except for *B. cousini*, which showed a unique value (Figure 5 and Figure 6).

The odds ratio of *B. glabrata* occurring at a sampling point was 28.8 times higher in the presence of *B. kuhniana* than in its absence (CI = 2.65–311.93; *p* = 0.0015; correctly classified = 86.96%; area under ROC curve = 70.87%), and 8.5 times higher in the presence of *S. marmorata* than in its absence (CI = 1.71–42.95; *p* = 0.0064; correctly classified = 88.43%; area under ROC curve = 73.87%). *Drepanotrema* (*p* = 0.1488) and *Pseudosuccinea* (*p* = 0.3222) were not significantly associated, and *B. tenagophila*, *B. straminea*, *B. cousini*, *Biomphalaria* sp., *Omalonyx,* and *Pomacea* were colinear with *B. glabrata*.

In addition, *B. glabrata* was the species with the vastest distribution and greatest abundance, followed by *S. marmorata* (Figure 1). Among these species, in Jaccard’s analysis, a similarity value of 0.75 was observed, compatible with the logistic regression analysis Appendix A and obtained odds ratio.

## 4. Discussion

The results represent the municipality’s first freshwater mollusk survey and thus provide data that contribute to the biomonitoring of the freshwater fauna. The variation in sampling efforts at some collection points disabled quantitative, seasonal comparisons about the richness and abundance of species. On the other hand, a variety of mollusk habitats were documented in Alvorada de Minas, where only medically important species had previously been documented.

In this scenario, there was a concurrent occurrence of two snail species, which has been discussed in the literature only to a small degree, e.g., a species of medical importance and an endangered species sharing the same environment. This demonstrates the need for extended malacological surveys to avoid inappropriate interventions, such as those used in the last century for the control of schistosomiasis, with the application of Niclosamide (2-amino ethanol salt of 2′, 5′-dichloro-4′-nitro salicylanilide; Bayluscide^®^, Bayer, Germany). The control of medically important mollusks has remained an essential component of integrated control programs, but chemical molluscicides are expensive, the application process is complex, and their use can generate major impacts and lethal effects on other organisms due to the lack of selective action [16,54,55,56,57]. In this respect, it is worth mentioning that specimens of *S. marmorata*, notified by ICMBio as an endangered species, would run the risk of having reduced populations within such interventions of snail control, as recently recommended by the WHO [58,59].

In Brazil, there are 15 species of threatened continental mollusks, classified in three categories (“Critically Endangered”, “Endangered”, or “Vulnerable”) and distributed in 12 Brazilian states, including the State of Minas Gerais, which has *S. marmorata* listed as a vulnerable species. However, this species received a NatureServe global heritage status rating of G5 (Least Concern). Some parts of their distribution coincide with protected areas [48,49,50,51]. Alvorada de Minas is not included in the Conservation Units, but is surrounded by more than five of them. The municipality is part of the “Central Corridor of Espinhaço”, a mountain range and preservation area of extremely high importance and priority for the conservation of the fauna [60].

In Article 3 of Law No. 5197 (Fauna Law) and for 37 articles of Law No. 9605 (Environmental Crimes Law) for Brazil, it is not a crime to kill wild animals, when it applies to animals harmful for agriculture or public health. However, in terms of a more global reflection of such interventions, the harmfulness of these animals must be characterized by a competent body or institution. In this respect, we believe that it is extremely important to indicate harmfulness regarding a restricted species and population of animals only in a certain region and not apply this to genera, families, or an order or class of animals in general.

These results highlight the importance of extensive malacological surveys for the understanding of mollusk diversity, their association with helminthic parasites, and ecological scenarios. The distribution patterns of limnic gastropods must be well established in order to precisely determine areas with species of medical importance, invasive species, and/or native species. The herein proposed methodology has the potential to be recommended for other qualitative studies of possible control interventions against intermediate hosts, and for monitoring or determination of conservation areas.

## Figures and Tables

**Figure 1 pathogens-11-01533-f001:**
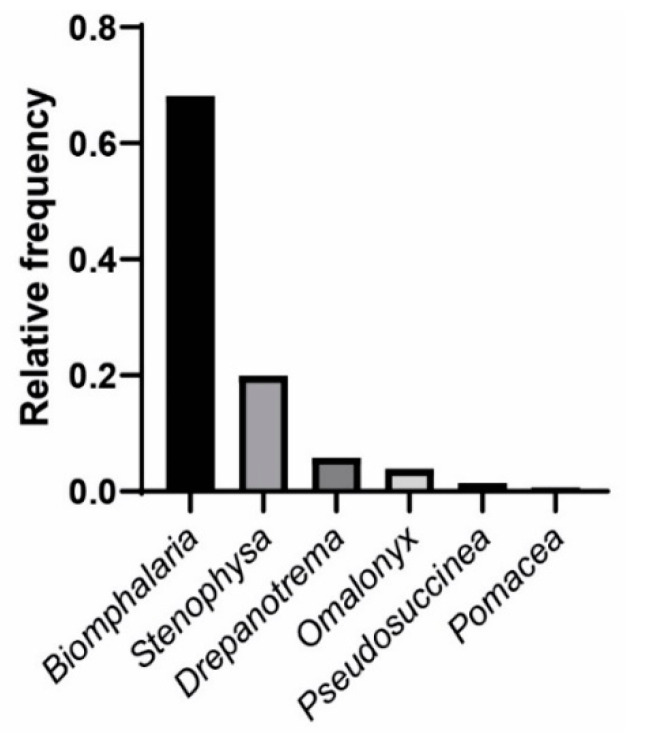
Relative frequency of collected and identified freshwater mollusk genera in Alvorada de Minas (MG, Brazil).

**Figure 2 pathogens-11-01533-f002:**
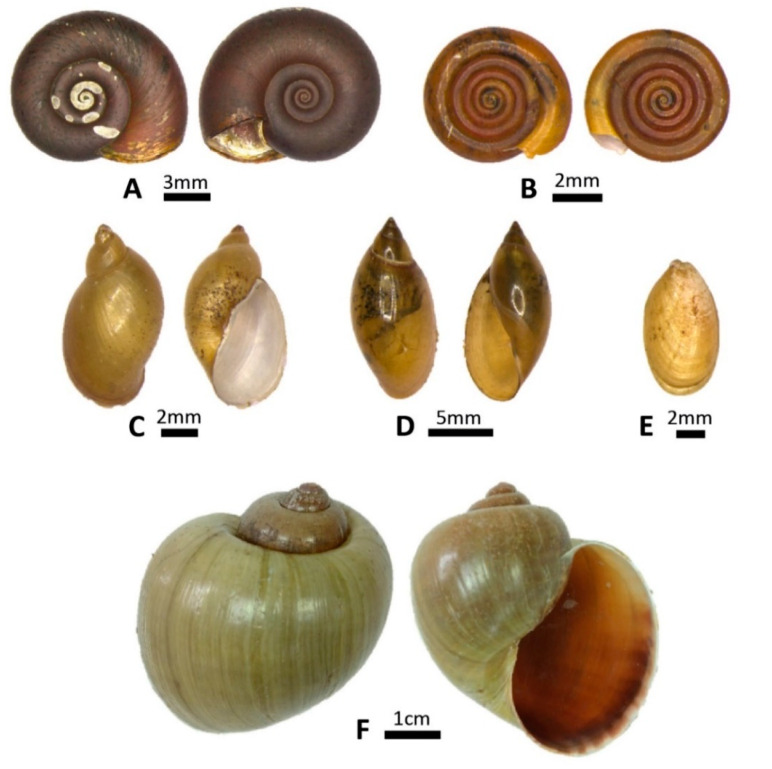
Representative photographs of shells of the freshwater malacofauna captured in the municipality of Alvorada de Minas (MG): (**A**) *Biomphalaria* sp.; (**B**) *Drepanotrema cimex*; (**C**) *Pseudosuccinea columella*; (**D**) *Stenophysa marmorata*; (**E**) *Omalonyx* sp., and (**F**) *Pomacea* sp. (photographed with LEICA DFC450).

**Figure 3 pathogens-11-01533-f003:**
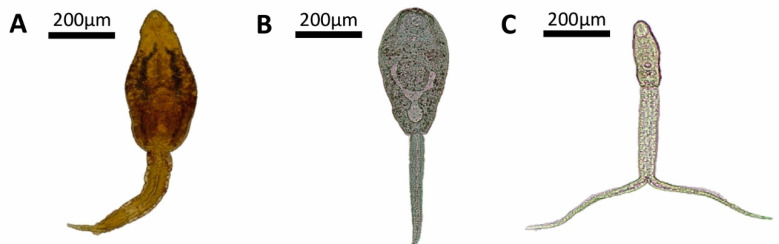
Cercariae of the types echinostome (**A**), Xiphidiocercaria (**B**) found in *Stenophysa marmorata*, and strigeocercaria (**C**) found in *Drepanotrema cimex* (photographed with LEICA ICC50 HD).

**Figure 4 pathogens-11-01533-f004:**
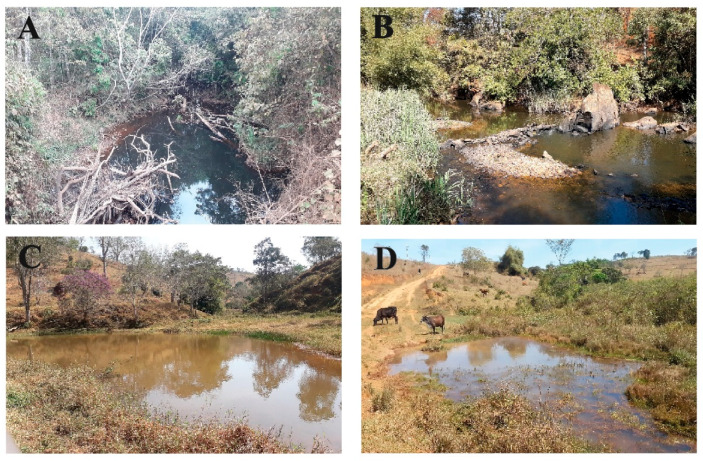
Examples of different freshwater mollusk habitats sampled in Alvorada de Minas: (**A**) small water hole, habitat with the highest number of sampled mollusk species (HU1); (**B**) Rio do Peixe river, lotic environment (HU4); (**C**) lake, located in the center of the town, used by the population for leisure activities (HU5); (**D**) artificial reservoir for cattle, habitat with the unique record of the species *B. cousini* (HU11).

**Figure 5 pathogens-11-01533-f005:**
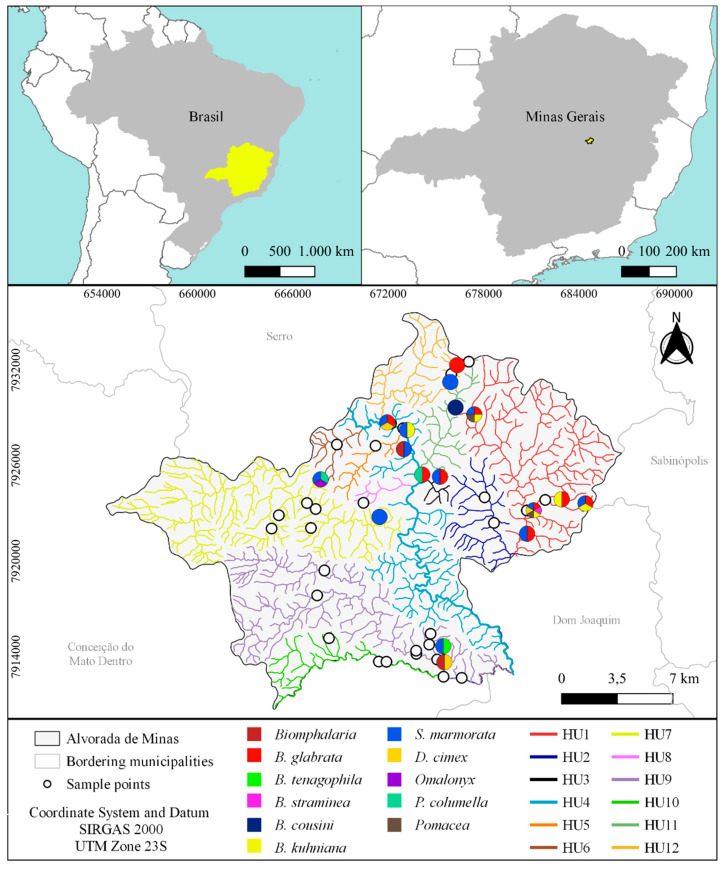
Localization of the State of Minas Gerais within Brazil and of the municipality Alvorada de Minas within Minas Gerais. Enlarged map of the municipality with the division of the hydrographic units (HU1–HU12) and distribution of the malacofauna (colored pies) in the municipality and according to the various collection points (o white spots) Appendix A.

**Figure 6 pathogens-11-01533-f006:**
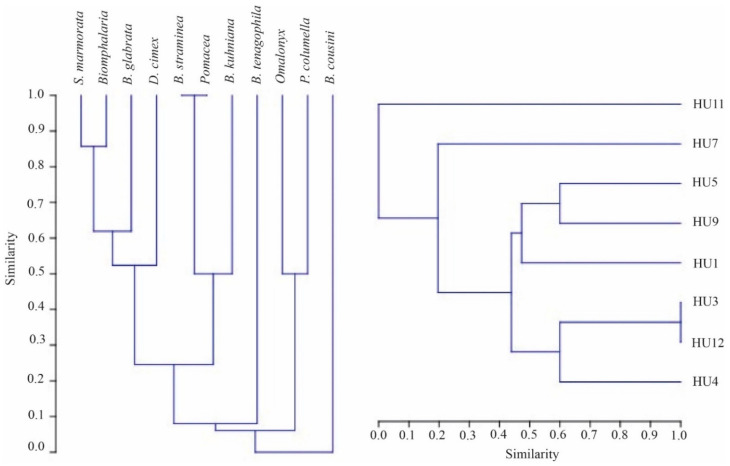
Bidirectional grouping analysis and similarity, based on the Jaccard similarity index for snail species (left diagram) and hydrographic units (right diagram; HU1, HU3, HU4, HU5, HU7, HU9, HU11, and HU12), according to Mueller-Dombois and Ellenberg (1974) and to Sneath and Sokal (1973).

**Table 1 pathogens-11-01533-t001:** Sampled limnic habitats and presence of mollusks in relation to lotic and lentic environments in Alvorada de Minas.

	Limnic Environments (Sampled Habitats)	Presence of Limnic Gastropods (Habitats with Mollusks)	*p*-Value
Lentic	29	15	0.02338
Lotic	16	1	
Total	45	16	

*p*-value obtained by the chi-square test.

## Data Availability

Not applicable.

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
