# Peer review of "Survey on Limnic Gastropods: Relationships between Human Health and Conservation"

_pathogens, 2022, doi:10.3390/pathogens11121533_

Round 1
Reviewer 1 Report
Small english revision is necessary.
1. What is the main question addressed by the research? malacologic data on fresh water snails in minas gerais brazil 2. Does it address a specific gap in the field? yes a lot of information is needed on malacology in brazil. 3. What does it add to the subject area compared with other published material? new information in the minas gerais state.
Author Response
- We thank Reviewer #1 for the positive evaluation of the submitted manuscript.
- We are sorry for typos and style issues and have now revised the entire manuscript one more time.
- The main goal of our manuscript is a detailed malacological survey on snails with and without medical or veterinary importance and a discussion on the necessity of control measures, since the WHO included snail control in their newly published guidelines, but without any considerations on ecological aspects. We believe that this is a very important point, which has to be considered in areas were conservation is an issue, especially in Brazil.

Reviewer 2 Report
This article presents a regional inventory of freshwater gastropod species from running and standing water bodies in a rural municipality in Brazil. A considerable number of snails has been sampled.
Frequences of snails/species in running and standing water were anaysed and correlations between these water units observed calculated.
The snails were also investigated for the presence of digenean larvae by cercarial release experiments.
The results showed richer gastropod faunas in standing waters and a high frequency and co-occurrence of two species, one transmitter of schistosomiasis, the other one an endangered species concerned by conservation laws. Additionally, the study recorded the presence of three different cercaria types, which were not closer classified/analysed.
In principle all data on the occurrence and distribution of organism on a regional or larger scale plus ecological background are valuable information and sharpen our insight in biological contexts.
But: the title of the manuscript is a mixture of vacuous first headline (Eclogical aspects and survey…) and promises a great thing in the second headline (relationships between human health and conservation). Indeed, the conflict between human needs and actions and conservation of biodiversity is a globally hot topic. And I agree that the conflict between conservation and vector control is worth much more attention.
But: this study provides some interesting data on freshwater gastropods and their distribution on a regional scale. It provides no data about threats caused by control measures and (to my knowledge) also does not give information about rare and endangered species in the region. The topic conflicts of control and conservation is not part of the methods and results. But it is the main topic treated in the discussion. In my opinion this study does not meet the title nor justify the discussion. The latter is poorly grounded on the local co-occurrence of one vector and one endangered species.
I would recommend to alter the title to what this study actually is: A regional inventory with some ecological aspects and to include the real results of the study in the discussion and to diminish the control/conserve discussion.
Anyway, it would be worth to consider a comprehensive review on the problematic of conflicts of medical vector control and its ecological consequences including appropriate examples and relevant literature.
70-84: this paragraph seems to be part of instructions to authors and not part of the manuscript
87: Gerais, on is part of the ancient Estrada
156-157: again part of instructions?
165: Omalonyx sp. (d´Orbigny, 1837)
175: schistosomiasis, [37] identified the species B. glabrata and B. straminea in Alvorada as intermediate hosts of S. mansoni??
185-186: occurrence of P. columella, an intermediate host species of F. hepatica was recorded first time??
209: the presence of larval stages identified as echinostome !
In Fig. 3 you name it genus Echinostoma . By what methods did you determine the genus?
210: cercariae, both eliminated shed by S. marmorata (infection rate: 0.89%).
215-216: What do you mean with estringeocercaria-type? Is it strideid?
Author Response
Reviewer 2
Comments and Suggestions for Authors
This article presents a regional inventory of freshwater gastropod species from running and standing water bodies in a rural municipality in Brazil. A considerable number of snails has been sampled.
Frequences of snails/species in running and standing water were anaysed and correlations between these water units observed calculated.
The snails were also investigated for the presence of digenean larvae by cercarial release experiments.
The results showed richer gastropod faunas in standing waters and a high frequency and co-occurrence of two species, one transmitter of schistosomiasis, the other one an endangered species concerned by conservation laws. Additionally, the study recorded the presence of three different cercaria types, which were not closer classified/analysed.
In principle all data on the occurrence and distribution of organism on a regional or larger scale plus ecological background are valuable information and sharpen our insight in biological contexts.
But: the title of the manuscript is a mixture of vacuous first headline (Ecological aspects and survey…) and promises a great thing in the second headline (relationships between human health and conservation). Indeed, the conflict between human needs and actions and conservation of biodiversity is a globally hot topic. And I agree that the conflict between conservation and vector control is worth much more attention.
But: this study provides some interesting data on freshwater gastropods and their distribution on a regional scale. It provides no data about threats caused by control measures and (to my knowledge) also does not give information about rare and endangered species in the region. The topic conflicts of control and conservation is not part of the methods and results. But it is the main topic treated in the discussion. In my opinion this study does not meet the title nor justify the discussion. The latter is poorly grounded on the local co-occurrence of one vector and one endangered species.
I would recommend to alter the title to what this study actually is: A regional inventory with some ecological aspects and to include the real results of the study in the discussion and to diminish the control/conserve discussion.
Anyway, it would be worth to consider a comprehensive review on the problematic of conflicts of medical vector control and its ecological consequences including appropriate examples and relevant literature.
- We thank Reviewer #2 for his assessment and recommendations for the submitted manuscript.
- Based on the revision, we modified the text according to some recommendations, reducing the emphasis on biological control and removing the focus from the ecological aspects of the title. Even though, we agree that the research is about a regional characterization of the malacofauna, we still believe that the study discusses an important issue, which has not yet been addressed in malacology, e.g. aspects on the relationship between human health and conservation.
- Although it is a regional study, the described pattern of distribution of molluscs might be identified in other endemic localities and through more detailed malacological surveys. Therefore, we recommend and encourage further research in this respect in order to better determine the potential risk for threatened species.
- The main goal of our manuscript is a detailed malacological survey on snails with and without medical or veterinary importance and a discussion on the necessity of control measures, since the WHO included snail control in their newly published guidelines, but without any considerations on ecological aspects. We believe that this is a very important point, which has to be considered in areas were conservation is an issue, especially in Brazil.
- To contextualize the status of S. marmorata as an endangered species, we have included the text below:
“In Brazil, there are 15 species of threatened continental molluscs classified in three categories ('Critically Endangered', 'Endangered' or 'Vulnerable') and distributed in 12 Brazilian states, including the State of Minas Gerais with S. marmorata as listed and vulnerable species. However, this species received a NatureServe global heritage status rating of G5 (Least Concern). In some parts of their distribution they coincide with protected areas [48-51]. Alvorada de Minas is not part of Conservation Units, but is surrounded by more than 5 of them. The municipality is part of the “Central Corridor of Espinhaço”, a mountain range and preservation area of extremely high importance and priority for the conservation of the fauna [60].
70-84: this paragraph seems to be part of instructions to authors and not part of the manuscript
- Thank you for the remark. The manuscript was revised again and the paragraphs referring to the instructions to authors part were removed.
87: Gerais, on is part of the ancient Estrada
- We changed the sentence to: ‘it is part of the "Estrada Real" tourist route’.
156-157: again part of instructions?
- A comment on this misconception was quoted above.
165: Omalonyx sp. (d´Orbigny, 1837)
- The scientific name was adapted to the taxonomic rules for the genus, Omalonyx d´Orbigny, 1837.
175: schistosomiasis, [37] identified the species B. glabrata and B. straminea in Alvorada as intermediate hosts of S. mansoni??
- Thanks for the suggestion, the modification has been made.
185-186: occurrence of P. columella, an intermediate host species of F. hepatica was recorded first time??
- Thanks for the suggestion, the modification has been made.
209: the presence of larval stages identified as echinostome !
In Fig. 3 you name it genus Echinostoma. By what methods did you determine the genus?
- Among the cercariae obtained, only the echinostoma-type cercariae had a more detailed morphological and molecular study. Among the 37-collar Echinostoma species ('revolutum' complex) our specimen resembled Echinostoma paraensei,, described for Brazil in the late 1960s and since then used as a biological model for various types of studies. However, the specimen in question has been identified as a cryptic species and that data has recently been submitted for publication (Pinto-H., personal communication).
- We included in the manuscript the reference used for the morphological identification for the genus:
Kostadinova A and Gibson DI (2000) The systematics of the echinostomes. In Fried B and Graczyk TK (eds) Echinostomes as Experimental Models for Biological Research. Dordrecht: Kluwer, 2000, pp. 31–57.
210: cercariae, both eliminated shed by S. marmorata (infection rate: 0.89%).
- Thanks for the suggestion, the modification has been made.
215-216: What do you mean with estringeocercaria-type? Is it strideid?
- Thanks for the suggestion, the modification has been made.

Reviewer 3 Report
All authors should be proof read this manuscript again, due to line 70 - 83 (material and method section) should not be appear in the text.
Moreover, some data are not coresponse together. The number of sample site in abstact (line20) , Method (line 90) and suplementary data should be corrected or give more information.
Figure2 the representative of snail shold be demoinstrated all species of Biophalaria spp. picture
Author Response
Reviewer 3
Comments and Suggestions for Authors
- We thank Reviewer #3 for the overall evaluation of the submitted manuscript.
All authors should be proof read this manuscript again, due to line 70 - 83 (material and method section) should not be appear in the text.
- Thank you for the remark. The manuscript was revised again and the paragraphs referring to the instructions to authors part were removed.
Moreover, some data are not coresponse together. The number of sample site in abstact (line20), Method (line 90) and suplementary data should be corrected or give more information.
- Thanks for the suggestion, the modification has been made.
Figure2 the representative of snail shold be demoinstrated all species of Biomphalaria spp. Picture
- Due to the lack of specimens of the photographic record of all species, we think it is more appropriate to represent only the genus. The lack of photographic record is due to the complete processing of soft tissues and also the snail shell for molecular biology (PCR-RFLP) because they are very small.
- The photographic record of the shells has been very important for the identification of the species Stenophysa marmorata, which in many scientific works has been mistakenly confused with the species Physella acuta (Draparnaud, 1805), the latter being an invasive species in Brazil.

Reviewer 4 Report
I thank the authors for their efforts of this work. It is an area of Brazil that clearly warrants ongoing surveillance of freshwater gastropods and their parasites. Below are my comments.
Line 28: S. marmorata needs to be italicized
Lines 40-84 and 156-158: I assume this was a journal requirement and will not be included in the final publication?
Line 111: During what time of day? The same time for each screening?
Line 253: Were water parameters taken at any of these habitats? It could be a reason why certain species group with one another. Also, the presence or absence of certain plant species.
Table S1: On this table, I would recommend adding a column of not only the GPS coordinate, but the name of the site or general area of where the snails were collected. It is a bit difficult to relate this to Figure 5 otherwise. The mini pie-charts indicate species collected in a general area and not at a specific location site, correct?
What is the history of Stenophysa vs Physa marmorata? Not only the history, but what do phylogenetic studies say about this species? I am a bit confused about the proper genus name, but the phylogenetic mitochondrial study would suggest Physa, but is also named differently in GenBank. I think a paragraph on this would be helpful to better understand the current validity and which genera the species belongs to based on molecular data, recent, and historical papers. If there is no consensus, that is also worth noting.
Author Response
Reviewer 4
Comments and Suggestions for Authors
I thank the authors for their efforts of this work. It is an area of Brazil that clearly warrants ongoing surveillance of freshwater gastropods and their parasites. Below are my comments.
- We thank Reviewer #4 for the overall positive evaluation of the submitted manuscript.
Line 28: S. marmorata needs to be italicized
- Thanks for the observation, the modification has been made.
Lines 40-84 and 156-158: I assume this was a journal requirement and will not be included in the final publication?
- Thank you for informing, the manuscript was all revised and the paragraphs referring to the instructions to authors part were removed.
Line 111: During what time of day? The same time for each screening?
- Due to a greater sampling of locations in the municipality, 46 points, collections were carried out in the morning and afternoon in 5 campaigns. A sampling effort of 20 minutes was carried out at each point, however the results are not subject to quantitative analysis.
Line 253: Were water parameters taken at any of these habitats? It could be a reason why certain species group with one another. Also, the presence or absence of certain plant species.
- The diversity of environments and the lack of adequate equipment did not allow a complex analysis of water parameters, such as: depth of 10 cm. The water quality parameter in the detection of fecal coliforms and descriptive characterization of environments was used to determine the risk areas published in the article: Coelho, P.R.S.; Ker, F.T.O.; Araújo, A.D.; Guimarães, R.J.P.S.; Negrão-Corrêa, D.A.; Caldeira, R.L.; Geiger, S.M. Identification of Risk Areas for Intestinal Schistosomiasis, Based on Malacological and Environmental Data and on Reported Human Cases. Med. 2021, 8, doi:10.3389/fmed.2021.642348.
Table S1: On this table, I would recommend adding a column of not only the GPS coordinate, but the name of the site or general area of where the snails were collected. It is a bit difficult to relate this to Figure 5 otherwise. The mini pie-charts indicate species collected in a general area and not at a specific location site, correct?
- We appreciate this observation, we have included a column with the hydrographic units in the supplementary table S1 to facilitate the identification of points. In Figure 5, sparklines represent specific locations and species by point. Although its location is slightly skewed by the program for better representation.
What is the history of Stenophysa vs Physa marmorata? Not only the history, but what do phylogenetic studies say about this species? I am a bit confused about the proper genus name, but the phylogenetic mitochondrial study would suggest Physa, but is also named differently in GenBank. I think a paragraph on this would be helpful to better understand the current validity and which genera the species belongs to based on molecular data, recent, and historical papers. If there is no consensus, that is also worth noting.
- Regarding gender, we follow Taylor's classification (2003), which presents a study of the morphology of the group. Although Wethington & Lydeard, 2007 also supports the use of the genus Stenophysa if it is included with Physinae rather than Aplexinae by molecular phylogeny. In Brazil, the most recent works, described by malacologist taxonomist researchers, have recommended and used this new classification as a consensus.
- Taylor, Dwight W. "Introduction to Physidae (Gastropoda: Hygrophila); biogeography, classification, morphology." Revista de biología tropical(2003): 1-287.
- Wethington, Amy R., and Charles Lydeard. "A molecular phylogeny of Physidae (Gastropoda: Basommatophora) based on mitochondrial DNA sequences." Journal of Molluscan Studies3 (2007): 241-257.
- Ximenes, Maria Eduarda Rocha, et al. "Para além das plantas: diversidade de moluscos límnicos no Instituto de Pesquisas Jardim Botânico do Rio de Janeiro." Pesquisa e Ensino em Ciências Exatas e da Natureza6 (2022): 3.
- Vendramini, Bianca Medeiros, and Eliane Pintor de Arruda. "Freshwater mollusk species of Itupararanga Reservoir, São Paulo, Brazil." Check List3 (2022): 629-708.
- Gonçalves, Isabela Cristina Brito, et al. "Moluscos de água doce da Floresta Nacional Mário Xavier, Seropédica, Rio de Janeiro." Pesquisa e Ensino em Ciências Exatas e da Natureza5 (2021): 3.
- Leal, Manuella F., et al. "Malacofauna of lotic environments in the Northeast and Brazilian semiarid region: current knowledge and new records." Anais da Academia Brasileira de Ciências93 (2021).
- Ohlweiler F, Takahashi F, Guimaraes M, Gomes S, Kawano T. Manual de gastrópodes límnicos e terrestres do estado de São Paulo associados às helmintoses. FAPESP, Porto Alegre, Brazil, 224 pp. (2010)

Round 2
Reviewer 2 Report
The authors have altered some passages and removed some inconsistencies in their manuscript. But, I am sorry to say, the main points are still those: The authors recorded a (low) number of (larger) snail species, did some “ecological” analyses about preference of lotic against lentic environments and about co-occurrences of species in sampling sites. Furthermore, they give some vague results of trematode findings in these gastropods. But nothing of these points is part of the discussion. Instead, they discuss about the conflict of conservation needs versus medical pest control. This is based on the (not surprising) co-occurrence of a protected species and schistosome intermediate host species.
The content of these article would be matched better with this title (e.g.):
Survey on limnic gastropods in the municipality Alvorada de Minas, Brazil: species inventory and comments on conflicts between pest control and nature conservation
But I do recommend, to divide these two topics into two manuscripts:
1) A local inventory of aquatic gastropods in the Alvorada de Minas, Brazil- this could be reshaped with more accuracy and published more locally or subject specific malacologically.
2) A review about the conflicting topics aquatic snail control versus nature conservation. This could include different aspects like chemical, biological and water management measures, influence of these actions on species other than the target species and on the environment, legal aspects as well as recommendation to improve the status quo. In my opinion this would be rather valuable and meritorious.